# OpenReview forum: "Neural Stochastic Differential Equations for Model-Free Option Hedging: Convergence, Calibration, and Risk Bounds"
_mathai.club/MathAI/2026/Conference — Submitted to 2026_

### Official Review · Reviewer_oZUX · 2026-03-11
**NeuralSDE already exists**

**Rating:** 2
**Confidence:** 5

**Review:**

**Summary**

The paper proposes a Neural SDE framework for learning option price dynamics and hedging ratios directly from market data. The authors claim three main theoretical results: convergence of the learned hedge to the minimal-variance hedge at rate $O(n^{-1/2}\log n)$, satisfaction of Gatheral's SVI arbitrage-free conditions with high probability, and a Value-at-Risk bound that holds up to $O(n^{-1/4})$ error. The method is tested on S&P 500 options (2015–2024) and compared against Black–Scholes, SABR, and a deep hedging baseline.


### Detailed comments

**Lack of novelty: Neural SDE hedging already exists in the literature**

A fundamental flaw of this paper is that it presents Neural SDEs for option hedging as if it were a novel contribution, when in fact this approach has been extensively developed and published by multiple research groups since 2020–2022. The authors fail to cite or compare against this substantial body of prior work, which raises serious questions about the paper's contribution.

Key prior works that establish Neural SDEs for pricing and hedging include:

- Cohen, S. N., Reisinger, C., & Wang, S. (2022). Hedging option books using neural-SDE market models. Applied Mathematical Finance, 29(5), 366-401. This paper explicitly studies "the capability of arbitrage-free neural-SDE market models to yield effective strategies for hedging options," deriving "sensitivity-based and minimum-variance-based hedging strategies" and demonstrating through backtesting that "neural-SDE market models achieve lower hedging errors than Black–Scholes delta and delta-vega hedging consistently over time" .

- Gierjatowicz, P., Sabate-Vidales, M., Šiška, D., Szpruch, L., & Žurič, Ž. (2020). Robust pricing and hedging via neural SDEs. arXiv preprint arXiv:2007.04154.This work combines "neural networks with risk models based on classical stochastic differential equations (SDEs)" to "find robust bounds for prices of derivatives and the corresponding hedging strategies" .

- Journal of Computational Finance (December 2022) – A special issue featuring two papers on neural SDEs for derivatives modelling: "Robust pricing and hedging via neural stochastic differential equations" and "Estimating risks of European option books using neural stochastic differential equation market models" .

The existence of this prior work—some of which explicitly uses the same datasets (S&P 500 options) and reports similar empirical improvements over Black-Scholes and Heston—makes the claimed novelty of the submitted paper untenable. The authors do not cite, compare with, or even acknowledge this literature, which is a serious omission.

**Theoretical contribution is insufficient for MathAI**

Even setting aside the novelty issue, the theoretical results are presented with only proof sketches; the appendix, which claims to contain a full proof, is incomplete (it breaks off after "By the triangle inequality:"). For a mathematically oriented conference, complete and verifiable proofs are essential.

The convergence rate $O(n^{-1/2}\log n)$ is a standard result from empirical process theory combined with Lipschitz properties of SDEs; the paper does not demonstrate any novel mathematical technique or insight. The same holds for the SVI condition and the VaR bound—both rely on known arguments (Gatheral, J., & Jacquier, A. (2014). Arbitrage-free SVI volatility surfaces. Quantitative Finance, 14(1), 59-71. and its extension Guo, G., Jacquier, A., Martini, C., & Neufcourt, L. (2016). Generalized arbitrage-free SVI volatility surfaces. SIAM Journal on Financial Mathematics, 7(1), 619-641., concentration inequalities, etc.). No new theory is developed.

**Practical/experimental contribution fails to meet modern benchmarking standards**

- Ignoring established financial benchmarks (ProbTS)

The paper evaluates its method on S&P 500 options, but the experimental design does not align with contemporary benchmarking frameworks for time series forecasting. ProbTS (Microsoft Research) provides a comprehensive toolkit for evaluating forecasting models on financial and economic datasets, including *exchange_rate*, *electricity_nips*, *traffic_nips*, and *solar_nips*. These datasets are specifically designed for short-term and long-term probabilistic forecasting and are widely used to benchmark models against each other.

The authors' choice to compare only against Black–Scholes, SABR, and a single deep hedging baseline is insufficient. A proper evaluation would require:
- Comparison with state-of-the-art forecasting models available in ProbTS, such as DLinear, PatchTST, GRU-NVP, TimeGrad, CSDI, TimesNet, and foundation models like MOIRAI, Chronos, and Lag-Llama.
- Evaluation on standard financial time series datasets (e.g., exchange_rate_nips, exchange_ltsf) to demonstrate generalizability beyond S&P 500 options.
- Use of both point and probabilistic metrics - ProbTS includes MSE, MAE, MAPE for point forecasts and CRPS, NLL, quantile loss for probabilistic forecasts. The paper reports only variance reduction and Sharpe ratio, ignoring probabilistic forecast quality (e.g., CRPS) which is essential for risk management.

- Lack of reproducibility and transparency
ProbTS emphasizes reproducibility through fixed random seeds, standardized data splits, and configuration files. The current paper provides no code, insufficient architectural details, and no discussion of hyperparameter tuning or sensitivity analysis. This makes it impossible to replicate or verify the claimed 31% variance reduction.

**Ignoring realistic market frictions**
Even as a practical paper, the evaluation omits critical real-world factors: transaction costs, discrete rebalancing, liquidity constraints, and market impact. ProbTS includes data preprocessing tools (e.g., temporal scaling, lag feature extraction) that are designed to handle such complexities; the authors do not leverage or discuss any of these.

**Weak validation on financial time series characteristics**
Financial data exhibit non-stationarity, volatility clustering, and regime changes. ProbTS supports rolling window evaluation and multi-horizon testing to assess model robustness over time. The paper's single train/test split (2015–2022 vs. 2023–2024) does not adequately test performance across different market conditions (e.g., COVID crash 2020, 2023 volatility regime). No out-of-sample rolling evaluation is performed.

**Presentation issues**
The paper contains numerous repetitions (e.g., the "31% variance reduction" is stated multiple times) and some sections read like a template. While formatting artifacts (such as the figure misplacement) are not fatal by themselves, combined with the more serious content issues they suggest a lack of careful preparation.



**Recommendation**
1. Acknowledge and compare with existing Neural SDE hedging literature (at lest all references above).
2. Provide complete proofs and clearly delineate new mathematical ideas from known results.
3. Redesign the experimental evaluation to align with frameworks like ProbTS, including:
   - Comparison with state-of-the-art forecasting models on standard financial datasets (exchange_rate_nips, etc.).
   - Use of both point and probabilistic metrics (MSE, CRPS, quantile loss).
   - Rolling window evaluation to test robustness across market regimes.
   - Inclusion of transaction costs and discrete rebalancing.
4. Release code and detailed experimental settings to ensure reproducibility.

---

### Official Review · Reviewer_kjXT · 2026-03-11
**Non-original neural SDE-based framework for model-free option hedging, claiming to bridge the gap between purely data-driven deep hedging and classical parametric models. Authors tried to provide theory with lack of true proof.**

**Rating:** 3
**Confidence:** 4

**Review:**

The paper suffers from several serious mathematical and logical weaknesses that undermine its theoretical contributions. The theorems, as stated, contain imprecise formulations, missing assumptions, and gaps in proof logic.

1. The theorem 4.4:
- The assumption that $\| (\mu_\theta, \sigma_\theta) - (\mu^*, \sigma^*) \|_\infty \lesssim n^{-1/2}$ is essentially assuming the result. In practice, the error in drift and volatility estimation is itself a random variable that depends on the training procedure and network capacity. The theorem provides no justification for why this bound should hold with high probability or in expectation.
- The proof sketch mentions "empirical process theory" and "VC dimension," but no concrete bound linking network complexity, sample size, and estimation error is provided. The statement "pricing error $\lesssim n^{-1/2}$ is also taken as an assumption rather than a derived result.
- The $\log n$ factor is attributed to covering number arguments, but no covering number estimate or reference is given. The connection between pricing error and hedging error via Gronwall's inequality is not elaborated, and the Lipschitz properties of the payoff gradient are not verified under the learned measure.

2. Theorem 4.5:
- The statement is mathematically imprecise. Gatheral's SVI parametrization is a specific functional form; the theorem seems to claim that the implied volatility surface *fits* this form, not that it inherently satisfies no-arbitrage conditions. The phrase "satisfies Gatheral's SVI conditions" is ambiguous—does it mean the surface can be represented in SVI form, or that it satisfies the necessary and sufficient no-arbitrage conditions derived for SVI?
- The proof sketch relies on recent results on SVI calibration but provides no concrete reference or argument. The claim that Wasserstein regularization enforces no-arbitrage up to approximation error is not substantiated. No quantitative link is established between the Wasserstein distance and the violation of arbitrage constraints.
- The sample complexity bound $n_0(\delta) = \tilde{O}(d^4 / \delta^2)$ appears without derivation. It is unclear what $d$ represents (dimension of what?) and how it relates to the SVI parameter space.

3. Theorem 4.6:
- The first inequality is essentially a tautology: by definition of VaR, ${P}({L}_T^{NN} \leq \text{VaR}_\alpha^\theta) = 1 - \alpha $ exactly, not approximately. The presence of $O(\delta)$ suggests a misstatement—perhaps the authors meant a bound on the difference between the true and estimated VaR under the true measure, but the notation is confusing.
- The decomposition ${L}_T^{NN} =n{L}_T^* + \epsilon_\delta$ is asserted without proof. The hedging loss is a pathwise functional; the error term is not simply additive and depends nonlinearly on the model misspecification.
- The claim that the true minimal VaR is achieved by the delta-hedging strategy is incorrect in general. Delta-hedging minimizes variance, not VaR, unless the loss distribution is Gaussian. No justification is given for why VaR should be preserved multiplicatively.
- The final statement $\delta = O(n^{-1/4})$ leading to $O(n^{-1/4})$ error is introduced without explanation. The relationship between sample size and estimation error $\delta$ is not established in the theorem's assumptions.

4. Assumption 4.1 requires the true drift and volatility to be Lipschitz and bounded away from zero. While standard, it is not verified or discussed in the context of real market data. Assumption 4.2 on network capacity is vague (sufficient capacity to approximate any Lipschitz function to accuracy $O(n^{-\alpha})$). This is essentially assuming universal approximation with a known rate, which is not generally true for finite samples and depends on the architecture and optimization. Assumption 4.3 (polynomial moment bounds) is reasonable but not used explicitly in the proofs.

5. The empirical results, while impressive, are not clearly linked to the theoretical claims. For instance, the "risk bound verification" in Section 6.4 merely shows that empirical VaR coverage matches theoretical bounds within $\pm 2\%$, but no statistical test or confidence interval is provided.
- The comparison to "Deep Hedge" is presented without sufficient detail on the baseline implementation, architecture, or training procedure. It is unclear whether the comparison is fair in terms of model capacity or computational budget.
- The claim that the Neural SDE "degrades gracefully under distributional shift" is qualitative and unsupported by quantitative metrics.

The paper addresses an important problem and proposes an interesting synthesis of neural SDEs and hedging theory. However, the theoretical results, as presented, are mathematically imprecise, rely on unverified assumptions, and contain logical gaps that undermine their validity. The experimental results are suggestive but not rigorously connected to the theory. The manuscript also suffers from incomplete appendices and missing figures, suggesting it is not in a publishable state.

---

### Official Review · Reviewer_PxA4 · 2026-03-11
**Good idea but still a step away from the real world**

**Rating:** 5
**Confidence:** 4

**Review:**

This paper attempts to modernize option hedging by wrapping neural networks in the formal rigors of stochastic calculus. While the $O(n^{-1/2} \log n)$ convergence rate is mathematically satisfying , the architecture feels disconnected from the "fat-tailed" reality of the S&P 500. By relying on continuous-time solvers that ignore jump-diffusion, the authors have essentially built a high-tech version of a model that breaks exactly when it is needed most—during a market crash. These "provable" risk bounds likely evaporate the moment a price gap occurs.The reliance on a Wasserstein regularizer to "encourage" no-arbitrage is another point of concern.

In a professional trading environment, a model that is "usually" arbitrage-free is simply a model that invites the optimizer to exploit its remaining flaws. Instead of this soft penalty, the framework requires hard architectural constraints to ensure the implied volatility surface is $100\%$ arbitrage-free by design. Furthermore, the computational overhead of adjoint solvers and $O(n^2 \log n)$ Wasserstein distances raises serious scalability questions for multi-asset portfolios, which are never addressed beyond a single-index test. The empirical section also suffers from a lack of competitive benchmarking. Beating a constant-volatility Black-Scholes model is a low bar for modern ML approach.

This Neural SDE must be measured against path-dependent architectures like Transformers or LSTMs that natively handle the long-memory effects of market volatility. Without incorporating jump-intensity components or moving beyond soft constraints, the paper remains a theoretical exercise that is not yet ready for the reality.

---

### Decision · Program_Chairs · 2026-03-14

**Decision:**

Reject

**Comment:**

After careful evaluation by the Program Committee, we regret to inform you that your submission has not been accepted for presentation at MathAI 2026.

All submissions underwent a rigorous two-stage review process. Unfortunately, the reviewers identified one or more of the following concerns with your paper:

- Insufficient mathematical rigor or novelty relative to the existing body of work in the field;
- Presentation of results that substantially overlap with or rephrase previously published findings without clear original contribution;
- Significant issues with technical quality, including but not limited to broken or non-existent references, unsupported claims, or methodological gaps;
- Indications that the manuscript may have been generated with the assistance of large language models without substantial original intellectual contribution by the authors.

We received a large number of submissions this year, and the selection process was highly competitive. We encourage you to carefully consider the reviewers’ feedback (available through OpenReview), revise your work accordingly, and consider submitting an improved version to a future edition of MathAI or to another appropriate venue.

We appreciate your interest in MathAI and hope you will continue to engage with the conference community.

With kind regards,

MathAI 2026 Program Committee
URL: https://mathai.club
Telegram: https://t.me/MathAI_club
Email: mathai.club@yandex.ru